# Nearest-Neighbor Sample Compression: Efficiency, Consistency, Infinite Dimensions

**Aryeh Kontorovich**
Department of Computer Science
Ben-Gurion University of the Negev
karyeh@cs.bgu.ac.il

**Sivan Sabato**
Department of Computer Science
Ben-Gurion University of the Negev
sabatos@bgu.ac.il

**Roi Weiss**
Department of Computer Science and Applied Mathematics
Weizmann Institute of Science
roiw@weizmann.ac.il

## Abstract

We examine the Bayes-consistency of a recently proposed 1-nearest-neighbor-based multiclass learning algorithm. This algorithm is derived from sample compression bounds and enjoys the statistical advantages of tight, fully empirical generalization bounds, as well as the algorithmic advantages of a faster runtime and memory savings. We prove that this algorithm is strongly Bayes-consistent in metric spaces with finite doubling dimension — the first consistency result for an efficient nearest-neighbor sample compression scheme. Rather surprisingly, we discover that this algorithm continues to be Bayes-consistent even in a certain infinite-dimensional setting, in which the basic measure-theoretic conditions on which classic consistency proofs hinge are violated. This is all the more surprising, since it is known that $k$-NN is not Bayes-consistent in this setting. We pose several challenging open problems for future research.

## 1 Introduction

This paper deals with Nearest-Neighbor (NN) learning algorithms in metric spaces. Initiated by Fix and Hodges in 1951 [16], this seemingly naive learning paradigm remains competitive against more sophisticated methods [8, 46] and, in its celebrated $k$-NN version, has been placed on a solid theoretical foundation [11, 44, 13, 47].

Although the classic 1-NN is well known to be inconsistent in general, in recent years a series of papers has presented variations on the theme of a *regularized* 1-NN classifier, as an alternative to the Bayes-consistent $k$-NN. Gottlieb et al. [18] showed that approximate nearest neighbor search can act as a regularizer, actually improving generalization performance rather than just injecting noise. In a follow-up work, [27] showed that applying Structural Risk Minimization to (essentially) the margin-regularized data-dependent bound in [18] yields a strongly Bayes-consistent 1-NN classifier. A further development has seen margin-based regularization analyzed through the lens of sample compression: a near-optimal nearest neighbor condensing algorithm was presented [20] and later extended to cover semimetric spaces [21]; an activized version also appeared [25]. As detailed in [27], margin-regularized 1-NN methods enjoy a number of statistical and computational advantages over the traditional $k$-NN classifier. Salient among these are explicit data-dependent generalization bounds, and considerable runtime and memory savings. Sample compression affords additional advantages, in the form of tighter generalization bounds and increased efficiency in time and space.

In this work we study the Bayes-consistency of a compression-based 1-NN multiclass learning algorithm, in both finite-dimensional and infinite-dimensional metric spaces. The algorithm is essentially the passive component of the active learner proposed by Kontorovich, Sabato, and Urner in [25], and we refer to it in the sequel as KSU; for completeness, we present it here in full (Alg. 1). We show that in finite-dimensional metric spaces, KSU is both computationally efficient and Bayes-consistent. This is the first compression-based multiclass 1-NN algorithm proven to possess both of these properties. We further exhibit a surprising phenomenon in infinite-dimensional spaces, where we construct a distribution for which KSU is Bayes-consistent while $k$-NN is not.

**Main results.** Our main contributions consist of analyzing the performance of KSU in finite and infinite dimensional settings, and comparing it to the classical $k$-NN learner. Our key findings are summarized below.

- In Theorem 2, we show that KSU is computationally efficient and strongly Bayes-consistent in metric spaces with a finite doubling dimension. This is the first (strong or otherwise) Bayes-consistency result for an efficient sample compression scheme for a multiclass (or even binary)[1] 1-NN algorithm. This result should be contrasted with the one in [27], where margin-based regularization was employed, but not compression; the proof techniques from [27] do not carry over to the compression-based scheme. Instead, novel arguments are required, as we discuss below. The new sample compression technique provides a Bayes-consistency proof for multiple (even countably many) labels; this is contrasted with the multiclass 1-NN algorithm in [28], which is not compression-based, and requires solving a minimum vertex cover problem, thereby imposing a 2-approximation factor whenever there are more than two labels.

- In Theorem 4, we make the surprising discovery that KSU continues to be Bayes-consistent in a certain infinite-dimensional setting, even though this setting violates the basic measure-theoretic conditions on which classic consistency proofs hinge, including Theorem 2. This is all the more surprising, since it is known that $k$-NN is not Bayes-consistent for this construction [9]. We are currently unaware of any separable[2] metric probability space on which KSU fails to be Bayes-consistent; this is posed as an intriguing open problem.

Our results indicate that in finite dimensions, an efficient, compression-based, Bayes-consistent multiclass 1-NN algorithm exists, and hence can be offered as an alternative to $k$-NN, which is well known to be Bayes-consistent in finite dimensions [12, 41]. In contrast, in infinite dimensions, our results show that the condition characterizing the Bayes-consistency of $k$-NN does not extend to all NN algorithms. It is an open problem to characterize the necessary and sufficient conditions for the existence of a Bayes-consistent NN-based algorithm in infinite dimensions.

**Related work.** Following the pioneering work of [11] on nearest-neighbor classification, it was shown by [13, 47, 14] that the $k$-NN classifier is strongly Bayes consistent in $\mathbb{R}^d$. These results made extensive use of the Euclidean structure of $\mathbb{R}^d$, but in [41] a weak Bayes-consistency result was shown for metric spaces with a bounded diameter and a bounded doubling dimension, and additional distributional smoothness assumptions. More recently, some of the classic results on $k$-NN risk decay rates were refined by [10] in an analysis that captures the interplay between the metric and the sampling distribution. The worst-case rates have an exponential dependence on the dimension (i.e., the so-called *curse of dimensionality*), and Pestov [33, 34] examines this phenomenon closely under various distributional and structural assumptions.

Consistency of NN-type algorithms in more general (and in particular infinite-dimensional) metric spaces was discussed in [1, 5, 6, 9, 30]. In [1, 9], characterizations of Bayes-consistency were given in terms of Besicovitch-type conditions (see Eq. (3)). In [1], a generalized "moving window" classification rule is used and additional regularity conditions on the regression function are imposed. The *filtering* technique (i.e., taking the first $d$ coordinates in some basis representation) was shown to be universally consistent in [5]. However, that algorithm suffers from the cost of cross-validating over *both* the dimension $d$ and number of neighbors $k$. Also, the technique is only applicable in

Hilbert spaces (as opposed to more general metric spaces) and provides only asymptotic consistency, without finite-sample bounds such as those provided by KSU. The insight of [5] is extended to the more general Banach spaces in [6] under various regularity assumptions.

None of the aforementioned generalization results for NN-based techniques are in the form of fully empirical, explicitly computable sample-dependent error bounds. Rather, they are stated in terms of the unknown Bayes-optimal rate, and some involve additional parameters quantifying the well-behavedness of the unknown distribution (see [27] for a detailed discussion). As such, these guarantees do not enable a practitioner to compute a numerical generalization error estimate for a given training sample, much less allow for a data-dependent selection of $k$, which must be tuned via cross-validation. The asymptotic expansions in [43, 37, 23, 40] likewise do not provide a computable finite-sample bound. The quest for such bounds was a key motivation behind the series of works [18, 28, 20], of which KSU [25] is the latest development.

The work of Devroye et al. [14, Theorem 21.2] has implications for 1-NN classifiers in $\mathbb{R}^d$ that are defined based on data-dependent majority-vote partitions of the space. It is shown that under some conditions, a fixed mapping from each sample size to a data-dependent partition rule induces a strongly Bayes-consistent algorithm. This result requires the partition rule to have a bounded VC dimension, and since this rule must be fixed in advance, the algorithm is not fully adaptive. Theorem 19.3 ibid. proves weak consistency for an inefficient compression-based algorithm, which selects among all the possible compression sets of a certain size, and maintains a certain rate of compression relative to the sample size. The generalizing power of sample compression was independently discovered by [31], and later elaborated upon by [22]. In the context of NN classification, [14] lists various condensing heuristics (which have no known performance guarantees) and leaves open the algorithmic question of how to minimize the empirical risk over all subsets of a given size.

The first compression-based 1-NN algorithm with provable optimality guarantees was given in [20]; it was based on constructing $\gamma$-nets in spaces with a finite doubling dimension. The compression size of this construction was shown to be nearly unimprovable by an efficient algorithm unless P=NP. With $\gamma$-nets as its algorithmic engine, KSU inherits this near-optimality. The compression-based 1-NN paradigm was later extended to semimetrics in [21], where it was shown to survive violations of the triangle inequality, while the hierarchy-based search methods that have become standard for metric spaces (such as [4, 18] and related approaches) all break down.

It was shown in [27] that a margin-regularized 1-NN learner (essentially, the one proposed in [18], which, unlike [20], did not involve sample compression) becomes strongly Bayes-consistent when the margin is chosen optimally in an explicitly prescribed sample-dependent fashion. The margin-based technique developed in [18] for the binary case was extended to multiclass in [28]. Since the algorithm relied on computing a minimum vertex cover, it was not possible to make it both computationally efficient and Bayes-consistent when the number of lables exceeds two. An additional improvement over [28] is that the generalization bounds presented there had an explicit (logarithmic) dependence on the number of labels, while our compression scheme extends seamlessly to countable label spaces.

**Paper outline.** After fixing the notation and setup in Sec. 2, in Sec. 3 we present KSU, the compression-based 1-NN algorithm we analyze in this work. Sec. 4 discusses our main contributions regarding KSU, together with some open problems. High-level proof sketches are given in Sec. 5 for the finite-dimensional case, and Sec. 6 for the infinite-dimensional case. Full detailed proofs can be found in [26].

## 2 Setting and Notation

Our instance space is the metric space $(\mathcal{X}, \rho)$, where $\mathcal{X}$ is the instance domain and $\rho$ is the metric. (See Appendix A in [26] for relevant background on metric measure spaces.) We consider a countable label space $\mathcal{Y}$. The unknown sampling distribution is a probability measure $\bar{\mu}$ over $\mathcal{X} \times \mathcal{Y}$, with marginal $\mu$ over $\mathcal{X}$. Denote by $(X, Y) \sim \bar{\mu}$ a pair drawn according to $\bar{\mu}$. The generalization error of a classifier $f : \mathcal{X} \to \mathcal{Y}$ is given by $\text{err}_{\bar{\mu}}(f) := \mathbb{P}_{\bar{\mu}}(Y \neq f(X))$, and its empirical error with respect to a labeled set $S' \subseteq \mathcal{X} \times \mathcal{Y}$ is given by $\widehat{\text{err}}(f, S') := \frac{1}{|S'|} \sum_{(x,y) \in S'} \mathbf{1}[y \neq f(x)]$. The optimal Bayes risk of $\bar{\mu}$ is $R^*_{\bar{\mu}} := \inf \text{err}_{\bar{\mu}}(f)$, where the infimum is taken over all measurable classifiers $f : \mathcal{X} \to \mathcal{Y}$. We say that $\bar{\mu}$ is *realizable* when $R^*_{\bar{\mu}} = 0$. We omit the overline in $\bar{\mu}$ in the sequel when there is no ambiguity.

For a finite labeled set $S \subseteq \mathcal{X} \times \mathcal{Y}$ and any $x \in \mathcal{X}$, let $X_{\mathrm{nn}}(x, S)$ be the nearest neighbor of $x$ with respect to $S$ and let $Y_{\mathrm{nn}}(x, S)$ be the nearest neighbor label of $x$ with respect to $S$:

$$(X_{\mathrm{nn}}(x, S), Y_{\mathrm{nn}}(x, S)) := \operatorname*{argmin}_{(x', y') \in S} \rho(x, x'),$$

where ties are broken arbitrarily. The 1-NN classifier induced by $S$ is denoted by $h_S(x) := Y_{\mathrm{nn}}(x, S)$. The set of points in $S$, denoted by $\boldsymbol{X} = \{X_1, \ldots, X_{|S|}\} \subseteq \mathcal{X}$, induces a *Voronoi partition* of $\mathcal{X}$, $\mathcal{V}(\boldsymbol{X}) := \{V_1(\boldsymbol{X}), \ldots, V_{|S|}(\boldsymbol{X})\}$, where each Voronoi cell is $V_i(\boldsymbol{X}) := \{x \in \mathcal{X} : \operatorname{argmin}_{j \in \{1, \ldots, |S|\}} \rho(x, X_j) = i\}$. By definition, $\forall x \in V_i(\boldsymbol{X})$, $h_S(x) = Y_i$.

A 1-NN algorithm is a mapping from an i.i.d. labeled sample $S_n \sim \bar{\mu}^n$ to a labeled set $S'_n \subseteq \mathcal{X} \times \mathcal{Y}$, yielding the 1-NN classifier $h_{S'_n}$. While the classic 1-NN algorithm sets $S'_n := S_n$, in this work we study a compression-based algorithm which sets $S'_n$ adaptively, as discussed further below.

A 1-NN algorithm is *strongly* Bayes-consistent on $\bar{\mu}$ if $\mathrm{err}(h_{S'_n})$ converges to $R^*$ almost surely, that is $\mathbb{P}[\lim_{n \to \infty} \mathrm{err}(h_{S'_n}) = R^*] = 1$. An algorithm is *weakly* Bayes-consistent on $\bar{\mu}$ if $\mathrm{err}(h_{S'_n})$ converges to $R^*$ in expectation, $\lim_{n \to \infty} \mathbb{E}[\mathrm{err}(h_{S'_n})] = R^*$. Obviously, the former implies the latter. We say that an algorithm is Bayes-consistent on a metric space if it is Bayes-consistent on all distributions in the metric space.

A convenient property that is used when studying the Bayes-consistency of algorithms in metric spaces is the *doubling dimension*. Denote the open ball of radius $r$ around $x$ by $B_r(x) := \{x' \in \mathcal{X} : \rho(x, x') < r\}$ and let $\bar{B}_r(x)$ denote the corresponding closed ball. The doubling dimension of a metric space $(\mathcal{X}, \rho)$ is defined as follows. Let $n$ be the smallest number such that every ball in $\mathcal{X}$ can be covered by $n$ balls of half its radius, where all balls are centered at points of $\mathcal{X}$. Formally,

$$n := \min\{n \in \mathbb{N} : \forall x \in \mathcal{X}, r > 0, \ \exists x_1, \ldots, x_n \in \mathcal{X} \text{ s.t. } B_r(x) \subseteq \cup_{i=1}^n B_{r/2}(x_i)\}.$$

Then the doubling dimension of $(\mathcal{X}, \rho)$ is defined by $\mathrm{ddim}(\mathcal{X}, \rho) := \log_2 n$.

For an integer $n$, let $[n] := \{1, \ldots, n\}$. Denote the set of all index vectors of length $d$ by $I_{n,d} := [n]^d$. Given a labeled set $S_n = (X_i, Y_i)_{i \in [n]}$ and any $\boldsymbol{i} = \{i_1, \ldots, i_d\} \in I_{n,d}$, denote the sub-sample of $S_n$ indexed by $\boldsymbol{i}$ by $S_n(\boldsymbol{i}) := \{(X_{i_1}, Y_{i_1}), \ldots, (X_{i_d}, Y_{i_d})\}$. Similarly, for a vector $\boldsymbol{Y}' = \{Y'_1, \ldots, Y'_d\} \in \mathcal{Y}^d$, denote by $S_n(\boldsymbol{i}, \boldsymbol{Y}') := \{(X_{i_1}, Y'_1), \ldots, (X_{i_d}, Y'_d)\}$, namely the sub-sample of $S_n$ as determined by $\boldsymbol{i}$ where the labels are replaced with $\boldsymbol{Y}'$. Lastly, for $\boldsymbol{i}, \boldsymbol{j} \in I_{n,d}$, we denote $S_n(\boldsymbol{i}; \boldsymbol{j}) := \{(X_{i_1}, Y_{j_1}), \ldots, (X_{i_d}, Y_{j_d})\}$.

## 3   1-NN majority-based compression

In this work we consider the 1-NN majority-based compression algorithm proposed in [25], which we refer to as KSU. This algorithm is based on constructing $\gamma$-*nets* at different scales; for $\gamma > 0$ and $A \subseteq \mathcal{X}$, a set $\boldsymbol{X} \subseteq A$ is said to be a $\gamma$-net of $A$ if $\forall a \in A, \exists x \in \boldsymbol{X} : \rho(a, x) \leq \gamma$ and for all $x \neq x' \in \boldsymbol{X}$, $\rho(x, x') > \gamma$.[3]

The algorithm (see Alg. 1) operates as follows. Given an input sample $S_n$, whose set of points is denoted $\boldsymbol{X}_n = \{X_1, \ldots, X_n\}$, KSU considers all possible scales $\gamma > 0$. For each such scale it constructs a $\gamma$-net of $\boldsymbol{X}_n$. Denote this $\gamma$-net by $\boldsymbol{X}(\gamma) := \{X_{i_1}, \ldots, X_{i_m}\}$, where $m \equiv m(\gamma)$ denotes its size and $\boldsymbol{i} \equiv \boldsymbol{i}(\gamma) := \{i_1, \ldots, i_m\} \in I_{n,m}$ denotes the indices selected from $S_n$ for this $\gamma$-net. For every such $\gamma$-net, the algorithm attaches the labels $\boldsymbol{Y}' \equiv \boldsymbol{Y}'(\gamma) \in \mathcal{Y}^m$, which are the empirical majority-vote labels in the respective Voronoi cells in the partition $\mathcal{V}(\boldsymbol{X}(\gamma)) = \{V_1, \ldots, V_m\}$. Formally, for $i \in [m]$,

$$Y'_i \in \operatorname*{argmax}_{y \in \mathcal{Y}} |\{j \in [n] \mid X_j \in V_i, Y_j = y\}|, \tag{1}$$

where ties are broken arbitrarily. This procedure creates a labeled set $S'_n(\gamma) := S_n(\boldsymbol{i}(\gamma), \boldsymbol{Y}'(\gamma))$ for every relevant $\gamma \in \{\rho(X_i, X_j) \mid i, j \in [n]\} \setminus \{0\}$. The algorithm then selects a single $\gamma$, denoted $\gamma^* \equiv \gamma^*_n$, and outputs $h_{S'_n(\gamma^*)}$. The scale $\gamma^*$ is selected so as to minimize a generalization error bound, which upper bounds $\mathrm{err}(S'_n(\gamma))$ with high probability. This error bound, denoted $Q$ in the algorithm, can be derived using a compression-based analysis, as described below.

**Algorithm 1** KSU: 1-NN compression-based algorithm

---

**Require:** Sample $S_n = (X_i, Y_i)_{i \in [n]}$, confidence $\delta$
**Ensure:** A 1-NN classifier
1: Let $\Gamma := \{\rho(X_i, X_j) \mid i, j \in [n]\} \setminus \{0\}$
2: **for** $\gamma \in \Gamma$ **do**
3:      Let $\boldsymbol{X}(\gamma)$ be a $\gamma$-net of $\{X_1, \ldots, X_n\}$
4:      Let $m(\gamma) := |\boldsymbol{X}(\gamma)|$
5:      For each $i \in [m(\gamma)]$, let $Y_i'$ be the majority label in $V_i(\boldsymbol{X}(\gamma))$ as defined in Eq. (1)
6:      Set $S_n'(\gamma) := (\boldsymbol{X}(\gamma), \boldsymbol{Y}'(\gamma))$
7: **end for**
8: Set $\alpha(\gamma) := \widehat{\text{err}}(h_{S_n'(\gamma)}, S_n)$
9: Find $\gamma_n^* \in \text{argmin}_{\gamma \in \Gamma} Q(n, \alpha(\gamma), 2m(\gamma), \delta)$, where $Q$ is, e.g., as in Eq. (2)
10: Set $S_n' := S_n'(\gamma_n^*)$
11: **return** $h_{S_n'}$

---

We say that a mapping $S_n \mapsto S_n'$ is a *compression scheme* if there is a function $\mathcal{C} : \cup_{m=0}^{\infty}(\mathcal{X} \times \mathcal{Y})^m \to 2^{\mathcal{X} \times \mathcal{Y}}$, from sub-samples to subsets of $\mathcal{X} \times \mathcal{Y}$, such that for every $S_n$ there exists an $m$ and a sequence $\boldsymbol{i} \in I_{n,m}$ such that $S_n' = \mathcal{C}(S_n(\boldsymbol{i}))$. Given a compression scheme $S_n \mapsto S_n'$ and a matching function $\mathcal{C}$, we say that a specific $S_n'$ is an $(\alpha, m)$-*compression* of a given $S_n$ if $S_n' = \mathcal{C}(S_n(\boldsymbol{i}))$ for some $\boldsymbol{i} \in I_{n,m}$ and $\widehat{\text{err}}(h_{S_n'}, S_n) \le \alpha$. The generalization power of compression was recognized by [17] and [22]. Specifically, it was shown in [21, Theorem 8] that if the mapping $S_n \mapsto S_n'$ is a compression scheme, then with probability at least $1 - \delta$, for any $S_n'$ which is an $(\alpha, m)$-compression of $S_n \sim \bar{\mu}^n$, we have (omitting the constants, explicitly provided therein, which do not affect our analysis)

$$\text{err}(h_{S_n'}) \le \frac{n}{n-m}\alpha + O(\frac{m\log(n) + \log(1/\delta)}{n-m}) + O(\sqrt{\frac{\frac{nm}{n-m}\alpha\log(n) + \log(1/\delta)}{n-m}}). \quad (2)$$

Defining $Q(n, \alpha, m, \delta)$ as the RHS of Eq. (2) provides KSU with a compression bound. The following proposition shows that KSU is a compression scheme, which enables us to use Eq. (2) with the appropriate substitution.[4]

**Proposition 1.** *The mapping $S_n \mapsto S_n'$ defined by Alg. 1 is a compression scheme whose output $S_n'$ is a $(\widehat{\text{err}}(h_{S_n'}), 2|S_n'|)$-compression of $S_n$.*

*Proof.* Define the function $\mathcal{C}$ by $\mathcal{C}((\bar{X}_i, \bar{Y}_i)_{i \in [2m]}) = (\bar{X}_i, \bar{Y}_{i+m})_{i \in [m]}$, and observe that for all $S_n$, we have $S_n' = \mathcal{C}(S_n(\boldsymbol{i}(\gamma); \boldsymbol{j}(\gamma)))$, where $\boldsymbol{i}(\gamma)$ is the $\gamma$-net index set as defined above, and $\boldsymbol{j}(\gamma) = \{j_1, \ldots, j_{m(\gamma)}\} \in I_{n,m(\gamma)}$ is some index vector such that $Y_i' = Y_{j_i}$ for every $i \in [m(\gamma)]$. Since $Y_i'$ is an empirical majority vote, clearly such a $\boldsymbol{j}$ exists. Under this scheme, the output $S_n'$ of this algorithm is a $(\widehat{\text{err}}(h_{S_n'}), 2|S_n'|)$-compression. $\square$

KSU is efficient, for any countable $\mathcal{Y}$. Indeed, Alg. 1 has a naive runtime complexity of $O(n^4)$, since $O(n^2)$ values of $\gamma$ are considered and a $\gamma$-net is constructed for each one in time $O(n^2)$ (see [20, Algorithm 1]). Improved runtimes can be obtained, e.g., using the methods in [29, 18]. In this work we focus on the Bayes-consistency of KSU, rather than optimize its computational complexity. Our Bayes-consistency results below hold for KSU, whenever the generalization bound $Q(n, \alpha, m, \delta_n)$ satisfies the following properties:

**Property 1** For any integer $n$ and $\delta \in (0, 1)$, with probability $1 - \delta$ over the i.i.d. random sample $S_n \sim \bar{\mu}^n$, for all $\alpha \in [0, 1]$ and $m \in [n]$: If $S_n'$ is an $(\alpha, m)$-compression of $S_n$, then $\text{err}(h_{S_n'}) \le Q(n, \alpha, m, \delta)$.

**Property 2** $Q$ is monotonically increasing in $\alpha$ and in $m$.

**Property 3** There is a sequence $\{\delta_n\}_{n=1}^{\infty}$, $\delta_n \in (0, 1)$ such that $\sum_{n=1}^{\infty} \delta_n < \infty$ and for all $m$,

$$\lim_{n \to \infty} \sup_{\alpha \in [0,1]} (Q(n, \alpha, m, \delta_n) - \alpha) = 0.$$

The compression bound in Eq. (2) clearly satisfies these properties. Note that Property 3 is satisfied by Eq. (2) using any convergent series $\sum_{n=1}^{\infty} \delta_n < \infty$ such that $\delta_n = e^{-o(n)}$; in particular, the decay of $\delta_n$ cannot be too rapid.

## 4   Main results

In this section we describe our main results. The proofs appear in subsequent sections. First, we show that KSU is Bayes-consistent if the instance space has a finite doubling dimension. This contrasts with classical 1-NN, which is only Bayes-consistent if the distribution is realizable.

**Theorem 2.** *Let $(\mathcal{X}, \rho)$ be a metric space with a finite doubling-dimension. Let Q be a generalization bound that satisfies Properties* 1-3*, and let $\delta_n$ be as stipulated by Property* 3 *for Q. If the input confidence $\delta$ for input size $n$ is set to $\delta_n$, then the 1-NN classifier $h_{S'_n(\gamma_n^*)}$ calculated by* KSU *is strongly Bayes consistent on $(\mathcal{X}, \rho)$: $\mathbb{P}(\lim_{n \to \infty} \mathrm{err}(h_{S'_n}) = R^*) = 1$.*

The proof, provided in Sec. 5, closely follows the line of reasoning in [27], where the strong Bayes-consistency of an adaptive margin-regularized 1-NN algorithm was proved, but with several crucial differences. In particular, the generalization bounds used by KSU are purely compression-based, as opposed to the Rademacher-based generalization bounds used in [27]. The former can be much tighter in practice and guarantee Bayes-consistency of KSU even for countably many labels. This however requires novel technical arguments, which are discussed in detail in Appendix B.1 in [26]. Moreover, since the compression-based bounds do not explicitly depend on ddim, they can be used even when ddim is infinite, as we do in Theorem 4 below. To underscore the subtle nature of Bayes-consistency, we note that the proof technique given here does not carry to an earlier algorithm, suggested in [20, Theorem 4], which also uses $\gamma$-nets. It is an open question whether the latter is Bayes-consistent.

Next, we study Bayes-consistency of KSU in infinite dimensions (i.e., with ddim $= \infty$) — in particular, in a setting where $k$-NN was shown by [9] not to be Bayes-consistent. Indeed, a straightforward application of [9, Lemma A.1] yields the following result.

**Theorem 3** (Cérou and Guyader [9])**.** *There exists an infinite dimensional separable metric space $(\mathcal{X}, \rho)$ and a realizable distribution $\bar{\mu}$ over $\mathcal{X} \times \{0, 1\}$ such that no $k_n$-NN learner satisfying $k_n/n \to 0$ when $n \to \infty$ is Bayes-consistent under $\bar{\mu}$. In particular, this holds for any space and realizable distribution $\bar{\mu}$ that satisfy the following condition: The set C of points labeled 1 by $\bar{\mu}$ satisfies*

$$\mu(C) > 0 \qquad and \qquad \forall x \in C, \quad \lim_{r \to 0} \frac{\mu(C \cap \bar{B}_r(x))}{\mu(\bar{B}_r(x))} = 0. \tag{3}$$

Since $\mu(C) > 0$, Eq. (3) constitutes a violation of the Besicovitch covering property. In doubling spaces, the Besicovitch covering theorem precludes such a violation [15]. In contrast, as [35, 36] show, in infinite-dimensional spaces this violation can in fact occur. Moreover, this is not an isolated pathology, as this property is shared by Gaussian Hilbert spaces [45].

At first sight, Eq. (3) might appear to thwart any 1-NN algorithm applied to such a distribution. However, the following result shows that this is not the case: KSU is Bayes-consistent on a distribution with this property.

**Theorem 4.** *There is a metric space equipped with a realizable distribution for which* KSU *is weakly Bayes-consistent, while any $k$-NN classifier necessarily is not.*

The proof relies on a classic construction of Preiss [35] which satisfies Eq. (3). We show that the structure of the construction, combined with the packing and covering properties of $\gamma$-nets, imply that the majority-vote classifier induced by *any* $\gamma$-net with a sufficienlty small $\gamma$ approaches the Bayes error. To contrast with Theorem 4, we next show that on the same construction, not all majority-vote Voronoi partitions succeed. Indeed, if the packing property of $\gamma$-nets is relaxed, partition sequences obstructing Bayes-consistency exist.

**Theorem 5.** *For the example constructed in Theorem 4, there exists a sequence of Voronoi partitions with a vanishing diameter such that the induced true majority-vote classifiers are not Bayes consistent.*

The above result also stands in contrast to [14, Theorem 21.2], showing that, unlike in finite dimensions, the partitions' vanishing diameter is insufficient to establish consistency when ddim $= \infty$. We conclude the main results by posing intriguing open problems.

**Open problem 1.** Does there exist a metric probability space on which some $k$-NN algorithm is consistent while KSU is not? Does there exist *any* separable metric space on which KSU fails?

**Open problem 2.** Cérou and Guyader [9] distill a certain Besicovitch condition which is necessary and sufficient for $k$-NN to be Bayes-consistent in a metric space. Our Theorem 4 shows that the Besicovitch condition is *not* necessary for KSU to be Bayes-consistent. Is it sufficient? What is a necessary condition?

## 5 Bayes-consistency of KSU in finite dimensions

In this section we give a high-level proof of Theorem 2, showing that KSU is strongly Bayes-consistent in finite-dimensional metric spaces. A fully detailed proof is given in Appendix B in [26].

Recall the optimal empirical error $\alpha_n^* \equiv \alpha(\gamma_n^*)$ and the optimal compression size $m_n^* \equiv m(\gamma_n^*)$ as computed by KSU. As shown in Proposition 1, the sub-sample $S_n'(\gamma_n^*)$ is an $(\alpha_n^*, 2m_n^*)$-compression of $S_n$. Abbreviate the compression-based generalization bound used in KSU by

$$Q_n(\alpha, m) := Q(n, \alpha, 2m, \delta_n).$$

To show Bayes-consistency, we start by a standard decomposition of the excess error over the optimal Bayes into two terms:

$$\mathrm{err}(h_{S_n'(\gamma_n^*)}) - R^* = \left(\mathrm{err}(h_{S_n'(\gamma_n^*)}) - Q_n(\alpha_n^*, m_n^*)\right) + \left(Q_n(\alpha_n^*, m_n^*) - R^*\right) =: T_I(n) + T_{II}(n),$$

and show that each term decays to zero with probability one. For the first term, Property 1 for $Q$, together with the Borel-Cantelli lemma, readily imply $\limsup_{n\to\infty} T_I(n) \leq 0$ with probability one. The main challenge is showing that $\limsup_{n\to\infty} T_{II}(n) \leq 0$ with probability one. We do so in several stages:

1. Loosely speaking, we first show (Lemma 10) that the Bayes error $R^*$ can be well approximated using 1-NN classifiers defined by the *true* (as opposed to empirical) majority-vote labels over fine partitions of $\mathcal{X}$. In particular, this holds for any partition induced by a $\gamma$-net of $\mathcal{X}$ with a sufficiently small $\gamma > 0$. This approximation guarantee relies on the fact that in finite-dimensional spaces, the class of continuous functions with compact support is dense in $L_1(\mu)$ (Lemma 9).

2. Fix $\tilde{\gamma} > 0$ sufficiently small such that any true majority-vote classifier induced by a $\tilde{\gamma}$-net has a true error close to $R^*$, as guaranteed by stage 1. Since for bounded subsets of finite-dimensional spaces the size of any $\gamma$-net is finite, the empirical error of any majority-vote $\gamma$-net almost surely converges to its true majority-vote error as the sample size $n \to \infty$. Let $n(\tilde{\gamma})$ sufficiently large such that $Q_{n(\tilde{\gamma})}(\alpha(\tilde{\gamma}), m(\tilde{\gamma}))$ as computed by KSU for a sample of size $n(\tilde{\gamma})$ is a reliable estimate for the true error of $h_{S_{n(\tilde{\gamma})}'(\tilde{\gamma})}$.

3. Let $\tilde{\gamma}$ and $n(\tilde{\gamma})$ be as in stage 2. Given a sample of size $n = n(\tilde{\gamma})$, recall that KSU selects an optimal $\gamma^*$ such that $Q_n(\alpha(\gamma), m(\gamma))$ is minimized over all $\gamma > 0$. For margins $\gamma \ll \tilde{\gamma}$, which are prone to over-fitting, $Q_n(\alpha(\gamma), m(\gamma))$ is not a reliable estimate for $h_{S_n'(\gamma)}$ since compression may not yet taken place for samples of size $n$. Nevertheless, these margins are discarded by KSU due to the penalty term in $Q$. On the other hand, for $\gamma$-nets with margin $\gamma \gg \tilde{\gamma}$, which are prone to under-fitting, the true error is well estimated by $Q_n(\alpha(\gamma), m(\gamma))$. It follows that KSU selects $\gamma_n^* \approx \tilde{\gamma}$ and $Q_n(\alpha_n^*, m_n^*) \approx R^*$, implying $\limsup_{n\to\infty} T_{II}(n) \leq 0$ with probability one.

As one can see, the assumption that $\mathcal{X}$ is finite-dimensional plays a major role in the proof. A simple argument shows that the family of continuous functions with compact support is no longer dense in $L_1$ in infinite-dimensional spaces. In addition, $\gamma$-nets of bounded subsets in infinite dimensional spaces need no longer be finite.

## 6 On Bayes-consistency of NN algorithms in infinite dimensions

In this section we study the Bayes-consistency properties of 1-NN algorithms on a classic infinite-dimensional construction of Preiss [35], which we describe below in detail. This construction was

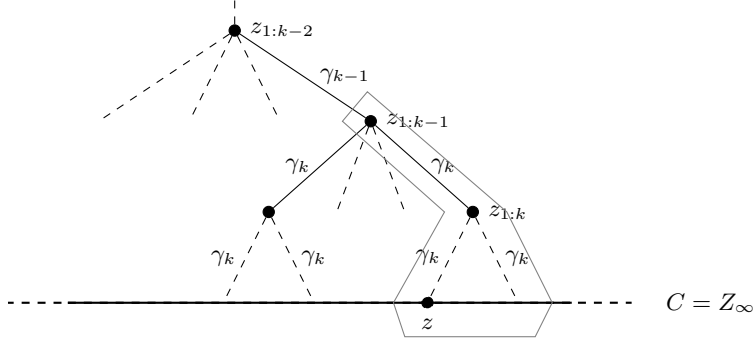

Figure 1: Preiss's construction. Encircled is the closed ball $\bar{B}_{\gamma_{k-1}}(z)$ for some $z \in C$.

first introduced as a concrete example showing that in infinite-dimensional spaces the Besicovich covering theorem [15] can be strongly violated, as manifested in Eq. (3).

**Example 1** (Preiss's construction)**.** *The construction (see Figure 1) defines an infinite-dimensional metric space $(\mathcal{X}, \rho)$ and a realizable measure $\bar{\mu}$ over $\mathcal{X} \times \mathcal{Y}$ with the binary label set $\mathcal{Y} = \{0, 1\}$. It relies on two sequences: a sequence of natural numbers $\{N_k\}_{k \in \mathbb{N}}$ and a sequence of positive numbers $\{a_k\}_{k \in \mathbb{N}}$. The two sequences should satisfy the following:*

$$\sum_{k=1}^{\infty} a_k N_1 \ldots N_k = 1; \quad \lim_{k \to \infty} a_k N_1 \ldots N_{k+1} = \infty; \quad and \quad \lim_{k \to \infty} N_k = \infty. \quad (4)$$

*These properties are satisfied, for instance, by setting $N_k := k!$ and $a_k := 2^{-k} / \prod_{i \in [k]} N_i$. Let $Z_0$ be the set of all finite sequences $(z_1, \ldots, z_k)_{k \in \mathbb{N}}$ of natural numbers such that $z_i \leq N_i$, and let $Z_\infty$ be the set of all infinite sequences $(z_1, z_2, \ldots)$ of natural numbers such that $z_i \leq N_i$.*

*Define the example space $\mathcal{X} := Z_0 \cup Z_\infty$ and denote $\gamma_k := 2^{-k}$, where $\gamma_\infty := 0$. The metric $\rho$ over $\mathcal{X}$ is defined as follows: for $x, y \in \mathcal{X}$, denote by $x \wedge y$ their longest common prefix. Then,*

$$\rho(x, y) = (\gamma_{|x \wedge y|} - \gamma_{|x|}) + (\gamma_{|x \wedge y|} - \gamma_{|y|}).$$

*It can be shown (see [35]) that $\rho(x, y)$ is a metric; in fact, it embeds isometrically into the square norm metric of a Hilbert space.*

*To define $\mu$, the marginal measure over $\mathcal{X}$, let $\nu_\infty$ be the uniform product distribution measure over $Z_\infty$, that is: for all $i \in \mathbb{N}$, each $z_i$ in the sequence $z = (z_1, z_2, \ldots) \in Z_\infty$ is independently drawn from a uniform distribution over $[N_i]$. Let $\nu_0$ be an atomic measure on $Z_0$ such that for all $z \in Z_0$, $\nu_0(z) = a_{|z|}$. Clearly, the first condition in Eq. (4) implies $\nu_0(Z_0) = 1$. Define the marginal probability measure $\mu$ over $\mathcal{X}$ by*

$$\forall A \subseteq Z_0 \cup Z_\infty, \quad \mu(A) := \alpha \nu_\infty(A) + (1 - \alpha)\nu_0(A).$$

*In words, an infinite sequence is drawn with probability $\alpha$ (and all such sequences are equally likely), or else a finite sequence is drawn (and all finite sequences of the same length are equally likely). Define the realizable distribution $\bar{\mu}$ over $\mathcal{X} \times \mathcal{Y}$ by setting the marginal over $\mathcal{X}$ to $\mu$, and by setting the label of $z \in Z_\infty$ to be 1 with probability 1 and the label of $z \in Z_0$ to be 0 with probability 1.*

As shown in [35], this construction satisfies Eq. (3) with $C = Z_\infty$ and $\mu(C) = \alpha > 0$. It follows from Theorem 3 that no $k$-NN algorithm is Bayes-consistent on it. In contrast, the following theorem shows that KSU is weakly Bayes-consistent on this distribution. Theorem 4 immediately follows from the this result.

**Theorem 6.** *Assume $(\mathcal{X}, \rho)$, $\mathcal{Y}$ and $\bar{\mu}$ as in Example 1.* KSU *is weakly Bayes-consistent on $\bar{\mu}$.*

The proof, provided in Appendix C in [26], first characterizes the Voronoi cells for which the true majority-vote yields a significant error for the cell (Lemma 15). In finite-dimensional spaces, the total measure of all such "bad" cells can be made arbitrarily close to zero by taking $\gamma$ to be sufficiently small, as shown in Lemma 10 of Theorem 2. However, it is not immediately clear whether this can be achieved for the infinite dimensional construction above.

Indeed, we expect such bad cells, due to the unintuitive property that for any $x \in C$, we have $\mu(\bar{B}_\gamma(x) \cap C)/\mu(\bar{B}_\gamma(x)) \to 0$ when $\gamma \to 0$, and yet $\mu(C) > 0$. Thus, if for example a significant

portion of the set $C$ (whose label is 1) is covered by Voronoi cells of the form $V = \bar{B}_\gamma(x)$ with $x \in C$, then for all sufficiently small $\gamma$, each one of these cells will have a true majority-vote 0. Thus a significant portion of $C$ would be misclassified. However, we show that by the structure of the construction, combined with the packing and covering properties of $\gamma$-nets, we have that in *any* $\gamma$-net, the total measure of all these "bad" cells goes to 0 when $\gamma \to 0$, thus yielding a consistent classifier.

Lastly, the following theorem shows that on the same construction above, when the Voronoi partitions are allowed to violate the packing property of $\gamma$-nets, Bayes-consistency does not necessarily hold. Theorem 5 immediately follows from the following result.

**Theorem 7.** *Assume $(\mathcal{X}, \rho)$, $\mathcal{Y}$ and $\bar{\mu}$ as in Example 1. There exists a sequence of Voronoi partitions $(\mathcal{P}_k)_{k \in \mathbb{N}}$ of $\mathcal{X}$ with $\max_{V \in \mathcal{P}_k} \operatorname{diam}(V) \leq \gamma_k$ such that the sequence of true majority-vote classifiers $(h_{\mathcal{P}_k})_{k \in \mathbb{N}}$ induced by these partitions is not Bayes consistent: $\liminf_{k \to \infty} \operatorname{err}(h_{\mathcal{P}_k}) = \alpha > 0$.*

The proof, provided in Appendix D, constructs a sequence of Voronoi partitions, where each partition $\mathcal{P}_k$ has *all* of its impure Voronoi cells (those with both 0 and 1 labels) being bad. In this case, $C$ is incorrectly classified by $h_{\mathcal{P}_k}$, yielding a significant error. Thus, in infinite-dimensional metric spaces, the shape of the Voronoi cells plays a fundamental role in the consistency of the partition.

**Acknowledgments.** We thank Frédéric Cérou for the numerous fruitful discussions and helpful feedback on an earlier draft. Aryeh Kontorovich was supported in part by the Israel Science Foundation (grant No. 755/15), Paypal and IBM. Sivan Sabato was supported in part by the Israel Science Foundation (grant No. 555/15).

## Footnotes

[1] An efficient sample compression algorithm was given in [20] for the binary case, but no Bayes-consistency guarantee is known for it.

[2] Cérou and Guyader [9] gave a simple example of a nonseparable metric on which all known nearest-neighbor methods, including $k$-NN and KSU, obviously fail.

[3] For technical reasons, having to do with the construction in Sec. 6, we depart slightly from the standard definition of a $\gamma$-net $\boldsymbol{X} \subseteq A$. The classic definition requires that (i) $\forall a \in A, \exists x \in \boldsymbol{X} : \rho(a, x) < \gamma$ and (ii) $\forall x \neq x' \in \boldsymbol{X} : \rho(x, x') \geq \gamma$. In our definition, the relations $<$ and $\geq$ in (i) and (ii) are replaced by $\leq$ and $>$.

[4] In [25] the analysis was based on compression with side information, and does not extend to infinite $\mathcal{Y}$.

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
