[Reviews · NeurIPS 2017]

Reviewer 1



The paper shows that Algo 1 is strongly Bayes-consistent in finite doubling-dimensional metric spaces. This is the second consistent nearest-neighbor variant (after the margin-regularised version in [23]). Interestingly here the Bayes consistency is obtained via a sample-compression scheme. Bayes consistency is then also proved in infinite dimensional setting. This seems to imply that Algo 1 is preferable not only to nearest neighbor but also to kNN. Pros: - Thorough theoretical analysis of a novel version of multi-class nearest neighbor learning (Algo 1), which builds on a number of recent results, but goes much deeper than existing work. - Algo 1 was derived by minimising a generalisation bound, so it is very nice to see that it actually has good properties. It shows an example where theory is useful in practice. - Well written - albeit necessarily terse, it is complete with definitions and background so it can be followed. Cons: - More a question than a con: Bayes-consistency is nice to have, but can anything be said about the finite sample behavior of Algo 1 ? (For instance it is known that the Tsybakov margin condition gives fast rates for the original kNN.)

Reviewer 2



This work develops a compression-based algorithm for multiclass learning; the authors claim the method is both efficient and strongly Bayes-consistent in spaces of finite doubling dimension. They also provide one example of a space of infinite doubling dimension with a particular measure for which their method is weakly Bayes consistent, whereas the same construction leads to inconsistency of k-NN rules. Overall, I think this paper is technically strong and seems to develop interesting results, but I have a few concerns about the significance of this paper which I will discuss below. If the authors can address these concerns, I would support this paper for acceptance. Detailed comments: I did not check the proofs in the appendix in detail but the main ideas appear to be correct. My main comments are on the efficiency of the method (which I expect the authors can easily address) and on the significant of the results in infinite dimensions. (I) I have a question about the efficiency of the algorithm: It is claimed that the complexity of Algorithm 1 is $O(n^4)$. I can see that the for loop traverses a set $\Gamma$ of cardinality at most $O(n^2)$, and in line 5 of the algorithm we pay at most $O(n)$ (which globally yields $O(n^3)$). What is the cost of constructing a $\gamma$-net? Is this $O(n^2)$, and if so, what is the algorithm for doing this? I do not yet believe the efficiency claim by the authors until this part is explained. It is absolutely necessary that Algorithm 1 is efficient, as otherwise one can just turn to standard $k$-NN methods (not based on compression) in order to obtain the same results, although through different methods. (II) Infinite dimensions: It is not clear to me why the results/discussion related to finite dimensions is significant. If I understand correctly, the results of Cérou and Guyader state that, for a particular (realizable) distribution, no $k$-NN learner (where $k$ grows with $n$ in the appropriate way) is Bayes-consistent. Thus, for a particular distribution, $k$-NN fails to be Bayes-consistent. But perhaps it succeeds for other realizable distributions satisfying equation (3). I can see that in the line after Theorem 3, the authors *seem to* claim that in fact the paper of Cérou and Guyader implies that *any* (realizable) distribution satisfying equation (3) will cause $k$-NN to be Bayes inconsistent; am I correct in this interpretation of Line 221? If so, is this easy to see from Cérou and Guyader's paper? On the other hand, the authors show that for a particular metric space of infinite doubling dimension and for a particular realizable distribution, KSU is Bayes-consistent. This is interesting provided that $k$-NN fails to be Bayes consistent for all constructions satisfying equation (3), but it is not interesting if the point is simply that, for the Preiss's construction, $k$-NN is not Bayes consistent while KSU is. It is therefore critical if the authors clarify the extent to which $k$-NN fails to be Bayes consistent (is it for all constructions satisfying equation (3) or just for Preiss's construction?). Also, the authors should mention Biau, Bunea, Wegkamp (2005), On the kernel rule for function classification'', because the paper of Cérou and Guyader (near the top of page 13 therein, or page 352 from internal numbering) claims that Biau et al. also show a nearest neighbor-type method that is weakly Bayes consistent in a space of infinite dimension. Mentioning this result (slightly) weakens the force of the authors' results in infinite dimensions. Minor comments: Lines 78-80: How do the authors know about the key motivation behind [15, 23, 16]? Line 277: I think $n(\tilde{\gamma})$ should be replaced by $n(\gamma)$. In the 3-point proof sketch of Theorem 2, from lines 262-281: The paper would benefit from a more explicit/longer explanation part 3. UPDATE AFTER REBUTTAL I am satisfied with the authors' responses and recommend acceptance. Also, regarding the confusion about the citations, I'm happy the authors were able to straighten out which paper was actually intended, while also pointing out the disadvantages of the approach from the older paper.

Reviewer 3



The authors consider a compressed 1NN algorithm that was introduced in a former article, and propose a model selection scheme for the compression level based on an upper bound of the generalization error of the classifier. The authors prove several theoretical results for the algorithm, both in the finite and infinite dimension setting. The article is well written and the sketches of proofs that are presented put the emphasis on the relevant steps of the proof without going too deep on technicalities. Overall the paper is interesting, my only concern is about the significance of the contribution for a wide audience. The computational cost of the procedure that is presented (O(n^4)) does not seem quite attractive, and after a quick look at the cited bibliography there does not seem to be many real applications of the strategy. Being not an expert of the field I would like the authors to motivate their procedure a little bit more. Is it only theoretical ? Can it be thought as a theoretical surrogate for any compression algorithm that could be routinely used in practice ? This would first widen a little bit the audience for the public, but that would also be consistent with the fact that in many places the authors mention that their procedure is attractive regarding its running time. Do this refer to the classification step once the compressed classifier is built ?